evolution, taxonomy and systematics, ecology

brush organ, insect phylogenetics, scent pad, scent patch, sexual selection

**Authors for correspondence:**
Wendy A. Valencia-Montoya
e-mail: wvalenciamontoya@g.harvard.edu
Naomi E. Pierce
e-mail: npierce@oeb.harvard.edu

[†]Joint first coauthors.
[‡]Joint senior coauthors.

# Evolutionary trade-offs between male secondary sexual traits revealed by a phylogeny of the hyperdiverse tribe Eumaeini (Lepidoptera: Lycaenidae)

Wendy A. Valencia-Montoya[1,†], Tiago B. Quental[1,2,†], João Filipe R. Tonini[1], Gerard Talavera[3], James D. Crall[1], Gerardo Lamas[4], Robert C. Busby[5], Ana Paula S. Carvalho[6], Ana B. Morais[7], Nicolás Oliveira Mega[8], Helena Piccoli Romanowski[8], Marjorie A. Liénard[9], Shayla Salzman[10], Melissa R. L. Whitaker[11], Akito Y. Kawahara[6], David J. Lohman[12,13,14], Robert K. Robbins[15,‡] and Naomi E. Pierce[1,‡]

[1]Department of Organismic and Evolutionary Biology and Museum of Comparative Zoology, Harvard University, Cambridge, MA 02138, USA
[2]Instituto de Biociências, Universidade de São Paulo, Brazil
[3]Institut Botànic de Barcelona (IBB, CSIC-Ajuntament de Barcelona), 08038 Barcelona, Catalonia, Spain
[4]Museo de Historia Natural, Universidad Nacional Mayor de San Marcos, Lima, Peru
[5]9275 Hollow Pine Drive, Estero, FL 34135, USA
[6]McGuire Center for Lepidoptera and Biodiversity, Florida Museum of Natural History, University of Florida, Gainesville, FL 32611, USA
[7]Departamento de Ecologia e Evolução, CCNE, Universidade Federal de Santa Maria, Santa Maria, RS, Brasil
[8]Departamento de Zoologia, Universidade Federal do Rio Grande do Sul, Porto Alegre, RS 91501970, Brazil
[9]Department of Biology, Lund University, Lund, Sweden
[10]School of Integrative Plant Sciences, Cornell University, Ithaca, NY 14853, USA
[11]Entomological Collection, Department of Environmental Systems Science, ETH Zürich, Zürich, Switzerland
[12]Biology Department, City College of New York, City University of New York, New York, NY 10031, USA
[13]PhD Program in Biology, Graduate Center, City University of New York, New York, NY 10016, USA
[14]Entomology Section, Zoology Division, Philippine National Museum of Natural History, Manila 1000, Philippines
[15]Department of Entomology, Smithsonian Institution, Washington, DC 20013-7012, USA

WAV-M, 0000-0001-9246-2330; JFRT, 0000-0002-4730-3805; GT, 0000-0003-1112-1345; GL, 0000-0002-3664-6730; MAL, 0000-0003-3193-3666; SS, 0000-0001-6808-7542; DJL, 0000-0002-0689-2906; RKR, 0000-0003-4137-786X; NEP, 0000-0003-3366-1625

Male butterflies in the hyperdiverse tribe Eumaeini possess an unusually complex and diverse repertoire of secondary sexual characteristics involved in pheromone production and dissemination. Maintaining multiple sexually selected traits is likely to be metabolically costly, potentially resulting in trade-offs in the evolution of male signals. However, a phylogenetic framework to test hypotheses regarding the evolution and maintenance of male sexual traits in Eumaeini has been lacking. Here, we infer a comprehensive, time-calibrated phylogeny from 379 loci for 187 species representing 91% of the 87 described genera. Eumaeini is a monophyletic group that originated in the late Oligocene and underwent rapid radiation in the Neotropics. We examined specimens of 818 of the 1096 described species (75%) and found that secondary sexual traits are present in males of 91% of the surveyed species. Scent pads and scent patches on the wings and brush organs associated with the genitalia were probably present in the common ancestor of Eumaeini and are widespread throughout the tribe. Brush organs and scent pads are negatively correlated across the phylogeny, exhibiting a trade-off in which lineages with brush organs are unlikely to regain scent pads and *vice versa*. In contrast, scent patches seem to facilitate the evolution

Proc. R. Soc. B 288: 20202512

of scent pads, although they are readily lost once scent pads have evolved. Our results illustrate the complex interplay between natural and sexual selection in the origin and maintenance of multiple male secondary sexual characteristics and highlight the potential role of sexual selection spurring diversification in this lineage.

# 1. Background

The lycaenid tribe Eumaeini represents one of the largest radiations of butterflies, with more than 950 species found in the New World and about 125 in the Palaearctic [1–4]. The tribe encompasses a staggering array of ecological and behavioural diversity, thriving in habitats from the Alaskan subarctic to temperate Chile, from sea level to the high Andes, and in environments as distinct as the Amazon rainforest and the Atacama Desert [5–7]. Species of Eumaeini possess a broad spectrum of adaptations to avoid predation, including complex larval interactions with ants, eyespots and tails that resemble a 'false head' at their posterior end, and aposematic coloration advertising toxins sequestered from their host plants [8–12].

The Eumaeini possesses the most complex and diverse suite of secondary sexual characters within Lycaenidae and possibly among all butterfly tribes [1]. Male secondary sexual organs in Eumaeini may occur on the head, wings, abdomen and/or legs [4,6,13–15]. These organs are involved in the generation and dissemination of pheromones important during courtship [13,16]. Three such structures occur widely in the Eumaeini: scent patches, scent pads and brush organs. Scent patches are common among Lepidoptera and are characterized by fused wing membranes [13,16,17]. Conversely, scent pads are unique to the Eumaeini and consist of intermembrane wing pockets with invaginations containing secretory cells [13,16]. Similarly, brush organs are mainly found in the Eumaeini and are scent-producing bundles of hollow setae with a chamber at the anterior end attached to an abdominal intersegmental membrane [13]. Several kinds of eumaeine male secondary sexual traits are restricted to a single genus or closely related genera and tend to occur in similar locations on the wings and the abdomen [13–15].

Multiple sexual characteristics are often costly to maintain, not only because they require the allocation of more metabolic resources, but also because they can lead to greater conspicuousness to predators [18–20]. Among a suite of males with multiple traits, selection will favour those males whose traits confer the greatest net fitness benefit because they are, for example, relatively less costly, more detectable, or more informative [21,22]. If each trait represents a major investment that is traded off against other life-history investments, it is likely that two such costly investments can induce an allocation conflict strong enough to drive a negative phenotypic correlation between the two traits [23,24]. At the microevolutionary level, there is widespread evidence of negative correlations between male traits; nonetheless, far less is known about trade-offs between male traits at macroevolutionary timescales [24]. Given the numerous male secondary organs exhibited by Eumaeini, this tribe is an ideal group to assess this hypothesis of trade-offs in the evolution of different male traits.

The foundation for understanding Eumaeini diversification and for integrating the evolution of secondary sexual traits along with the temporal history of the group is a comprehensive phylogeny. Eumaeini accounts for an estimated 5.7% of butterfly species, but a comprehensive phylogenetic hypothesis has not yet been proposed for the tribe. In this study, we use phylogenomic data to reconstruct evolutionary relationships within Eumaeini and to investigate patterns of evolution of secondary sexual characters. In particular, we map the secondary sexual traits onto the phylogeny, investigate whether the evolution of these male traits is co-dependent, and whether gains and losses of these traits occur at the same rate.

# 2. Methods

## (a) Taxon sampling, molecular methods and data cleaning

We collected and sequenced 187 species representing 79 (approx. 91%) of the 87 described Eumaeini genera [3], including members of all 15 taxonomic sections [3] (collection information and vouchers in the electronic supplementary material, table S3). For 61 of the sampled genera (approx. 77%), we were able to include at least two representative species. Based on a higher level phylogeny of butterflies [25], we included four outgroup species from closely related tribes (Arhopalini, Deudorigini and Tomarini).

We assembled two molecular datasets for phylogenetic analyses. First, a phylogenomic dataset with 74 species in 69 genera using a 450-loci anchored hybrid enrichment kit developed for butterflies (BUTTERFLY 1.0 [25]). We extracted DNA using an Omniprep Genomic DNA purification kit (G-Biosciences, St Louis, MO, USA). We only included loci that were captured in at least approximately 75% of the samples. Thus, only 378 of 450 possible loci were retained, amounting to up to 161 524 bp per sample. The backbone data matrix included 78 species (74 Eumaeini species and four outgroups), of which approximately 4.8% were missing data.

In addition to the 74 species with phylogenomic data, we gathered data for another 113 species using a 13-loci kit developed for butterflies [26] and five markers using standard Sanger sequencing. Thus, the full matrix for phylogenetic analyses included 191 species (187 eumaeines and four outgroups), 14 loci and 11 878 bp, of which approximately 12.2% were missing data. The list of the 379 sequenced loci used in the backbone and full data analysis is detailed in the data repository. Finally, we aligned each locus with MAFFT v. 7 [27] (see electronic supplementary material, Methods for more information on sampling and sequencing).

## (b) Phylogenetic analysis

We conducted phylogenetic reconstruction in two steps. First, we inferred a backbone tree using the phylogenomic dataset (378 loci, 78 spp.). We then used this backbone tree as a constraint for the phylogeny, including all samples (14 loci, 191spp.). We performed maximum-likelihood (ML) analyses on the concatenated matrices and used coalescent-based methods to account for gene tree incongruence. For ML analysis of the concatenated matrices, we used ModelFinder [28] for model selection and then found the best partitioning scheme using Partition Finder v. 2.0.0 [29]. We then incorporated the best partitioning scheme for phylogenetic inference in IQ-TREE v. 2.0 [30].

We used the variance among tree log-likelihoods and differences in tree topology as indicated by Robinson–Foulds distances as the criteria to identify the optimal tree. For the backbone dataset, 10 independent likelihood searches were sufficient to ensure convergence to a global optimum. We then selected the tree with the highest likelihood from these runs, and after collapsing nodes with bootstrap support of less than 80%, we used this tree as a constraint for the phylogenetic reconstruction using all samples.

Since fewer loci often lead to a more complex likelihood surface with multiple optima [31], we performed 500 independent runs with different values of perturbation strength. For the tree with the highest likelihood out of the 500 runs, we evaluated branch support with 10 000 ultrafast bootstrap replicates. We also performed 10 runs with the complete concatenated matrix (i.e. 14 loci, 191 spp.) without constraints and compared our results to the constrained tree.

In addition to the concatenation-based analysis, we also inferred species trees under the multispecies coalescent model [32] for the 378-loci dataset. We estimated the best partitioning schemes and substitution models in IQ-TREE v. 2.0, followed by phylogenetic reconstruction for each individual locus and support analysis with 1000 ultrafast bootstrap replicates. We collapsed nodes separated by short branches into polytomies in IQ-TREE v. 2.0 following [33]. We used the resulting ML gene trees as input for ASTRAL-II v. 4.10.8 [32] to build the species tree (electronic supplementary material, Methods).

## (c) Divergence time estimation and geographic patterns

We used the ML tree inferred with 191 species for estimating divergence times in MCMCtree v. 4.9 [34]. We performed a likelihood approximation with the calculation of the gradient and Hessian matrix of the branch lengths to speed computation. Given the large size of the dataset, we ran the concatenated alignment under the F84 substitution model and gamma with five rate parameters following [25].

Because fossils have not yet been found within Eumaeini or closely related tribes, we incorporated secondary node calibrations based only on recently published phylogenies [25,35] (electronic supplementary material, table S1). The remaining node age priors were set to uniform. Finally, to estimate ancestral ranges in the Eumaeini phylogeny, we used BioGeoBEARS as implemented in R [36] (electronic supplementary material, table S2 Methods).

## (d) Coding of secondary sexual characters

We carried out a survey of secondary sexual characters inspecting specimens from the Museum of Comparative Zoology (Cambridge, MA), the Florida Museum of Natural History, McGuire Center for Lepidoptera and Biodiversity (Gainesville, FL) and the Smithsonian Institution's National Museum of Natural History (Washington, DC). We examined at least 10 male specimens from 793 species to code the presence or absence of these male secondary sexual characteristics. For the remaining 25 species, we reviewed the literature to record the occurrence of male secondary sexual traits [5–7,13,15]. For this study, we focused on scent patches, scent pads and brush organs—structures that occur widely throughout the Eumaeini [13]. In addition to these widespread organs, we surveyed other androconial organs that are typically located in similar positions on the wings to assess the overall extent of secondary male characteristics, although no homology is assumed. These structures included: androconia located between the costal and subcostal vein on the dorsal forewing (CSbA), androconia located at the dorsal side of the hindwing (DHwA), androconia located at the central side of the forewing (VFwA) and androconia located at the ventral side of the hindwing (VHwA).

## (e) Analyses of phenotypic evolution

For the 187 species included in the phylogeny, we used a combination of ML state reconstruction methods and Bayesian estimation to test models of codependency and calculate rates of evolutionary transitions. We only conducted analyses for scent pads, scent patches and brush organs because these traits are widespread across the tribe and homology can be assumed.

We considered two matrices for trait evolution: (i) a single-parameter model that assumes reverse and forward transitions are equally probable (equal rates, ER) and (ii) a two-parameter model that allows different rates for forward and reverse transitions (all-rates-different, ARD). We tested which alternative model better fit our data using a likelihood ratio test and calculating $p$-values with a chi-square test. We then reconstructed ancestral states for brush organs, scent pads and scent patches by generating 1000 character histories using stochastic character mapping for the ultrametric tree under the best model as implemented in phytools [37].

We examined the correlated evolution of traits accounting for phylogenetic uncertainty by randomly sampling 1000 trees from the 10 000 bootstrap replicates. For ML analyses, we followed Pagel's approach [38] as implemented in the function 'fitPagel' from the R package phytools [37] and used the optimization method 'fitDiscrete' from the geiger package [39]. We tested two models for all possible combinations of scent patch, scent pad and brush organ: independent evolution and reciprocal dependence. We calculated AIC weights to select the best-fitting model. Using the best model, we then examined the distribution of likelihood ratios and $p$-values to assess the significance and recorded the transition rates for each tree. Adopting a Bayesian approach, we used BayesTraits v. 3.0.0 [40] to test for correlated evolution by comparing models fitted under 'Independent Discrete' and 'Dependent Discrete' [38]. We estimated rate posteriors using reversible jump MCMC and a hyperprior with exponential distribution (0–30) and ran the MCMC chain for 100 000 000 generations, discarding the first 1 000 000 as burn-in, and sampling every 5000th iteration. We ran 2000 sampling stones each with 50 000 iterations and assessed significance by computing Bayes factors between independent and dependent models. We calculated effective sample size (ESS) as the convergence diagnostic for the MCMC runs using the R package 'coda' [41].

## 3. Results

## (a) Phylogeny of Eumaeini

ML reconstruction yielded statistically well-supported trees that were largely consistent with the coalescent-based species tree (figure 1; electronic supplementary material, figures S1 and S3). The tribe Eumaeini was recovered as a monophyletic group with high support, regardless of the analysis framework or dataset (figure 1; electronic supplementary material, figures S1–S5). All inferred phylogenies recovered unambiguously an early divergence event leading to two major lineages within Eumaeini (figure 1; electronic supplementary material, figures S1–S5). One of these lineages is depicted as pink in figure 1 and includes some of the most species-rich genera within the tribe, and the only ones with a Palaearctic distribution, such as *Callophrys* and *Satyrium* (figure 1; electronic supplementary material, figures S8 and S9). The other lineage comprises the remaining genera (figure 1) and represents the bulk of Neotropical Eumaeini (figure 1; electronic supplementary material, figures S8 and S9). Notably, *Bistonina* and *Trichonis* are sister genera and the closest clade to all remaining Neotropical lineages (figure 1). Eight main clades were recovered (indicated by different colours in figure 1), all showing high support (bootstrap support >95%). Relations between three clades (red, blue and green in figure 1) exhibited substantial conflict between concatenation and coalescent results (electronic supplementary material, figures S1–S7), suggesting a relatively

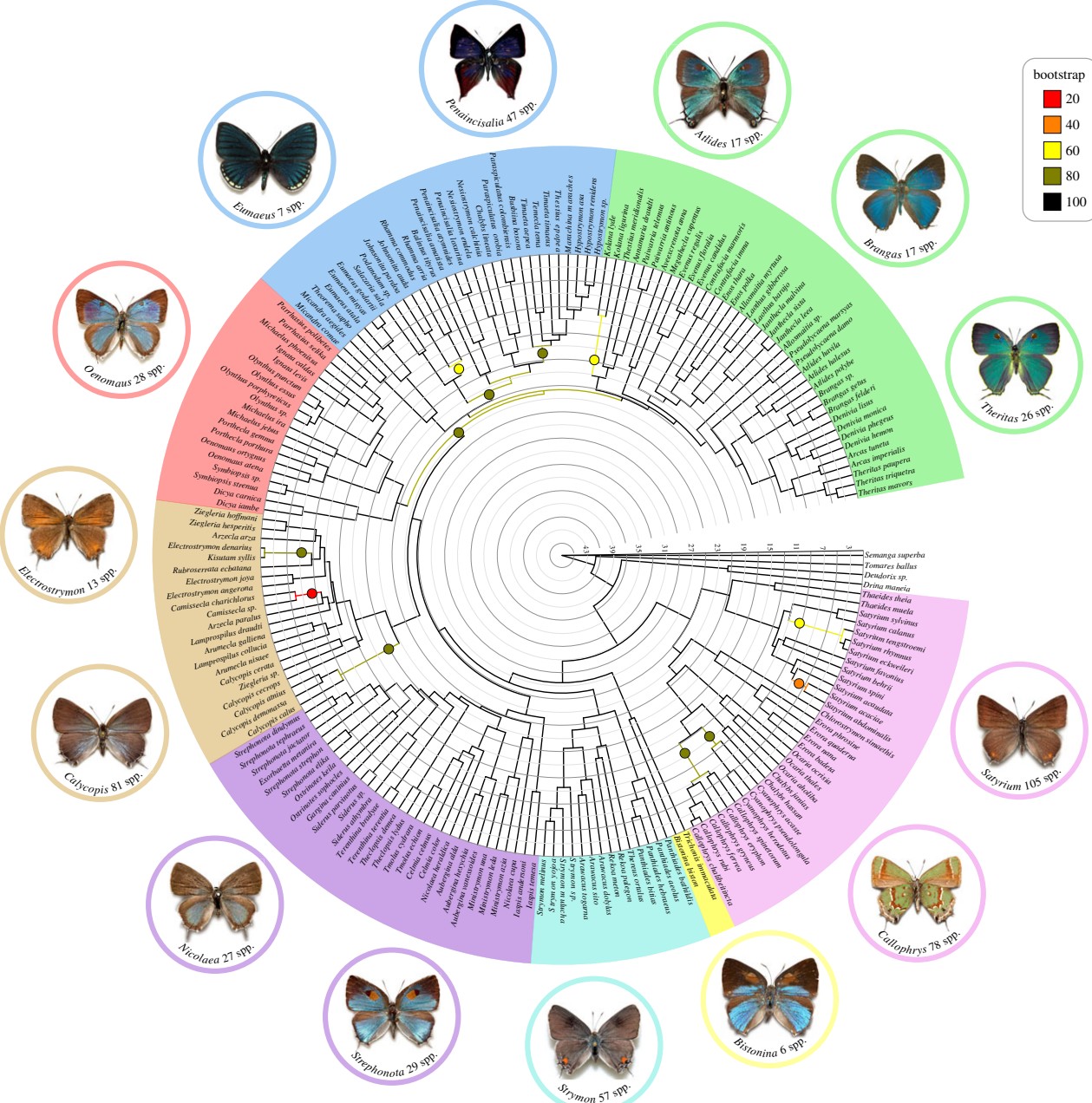

**Figure 1.** Dated phylogeny of Eumaeini based on molecular data and including 187 of the 1096 described species with representatives of 77 (90%) recognized genera. Consensus tree inferred using maximum likelihood and calibrated using approximate likelihood estimation and five secondary calibration points. The colour-coding of branches and circles at nodes (please refer to the online version for colour) indicates bootstrap support values as specified in the legend (circles only for branches with <100 support), and the scale represents millions of years ago (Ma). Colours of tip labels refer to the main clades recovered across different analyses. Images of butterflies were included for the most diverse genera within each group. (Online version in colour.)

larger impact of incomplete lineage sorting (ILS) or introgression. Of the 61 genera for which we had more than one representative species, 48 (approx. 80%) were supported as monophyletic, and 13 were found to be paraphyletic and/or polyphyletic (figure 1).

The most recent common ancestor of crown Eumaeini appeared in the Oligocene around 30 Ma (95% CI = 23.78–33.54 Ma). The main Neotropical and temperate clades evolved about 26.46 Ma (95% CI = 20.91–30.06 Ma) and 24.03 Ma (95% CI = 19.66–28.28 Ma), respectively (electronic supplementary material, figure S11). All seven main lineages recovered within the Neotropical clade originated together at roughly the same time in the Early Miocene, about 20 Ma (figure 1; electronic supplementary material, figure S11). The

biogeographic analysis lent support to a Dispersal-Extinction-Cladogenesis model [42]) with a founder-event jump dispersal parameter ($j$) (electronic supplementary material, table S2, although see [43]) that inferred the origin of the tribe in the Neotropics followed by dispersal to the Nearctic and Palaearctic by ancestors of the genera *Satyrium* and *Callophrys*.

## (b) Evolution of male secondary sexual characters

Approximately 91% of the 818 examined species have at least one androconial organ (figure 2). About 9% of Eumaeini species lack androconial organs (9.16%, N = 75; figure 2b,c), and roughly half have only one (49.45%, N = 405; figure 2b, c). Of the remaining species, 28.44% have two androconial

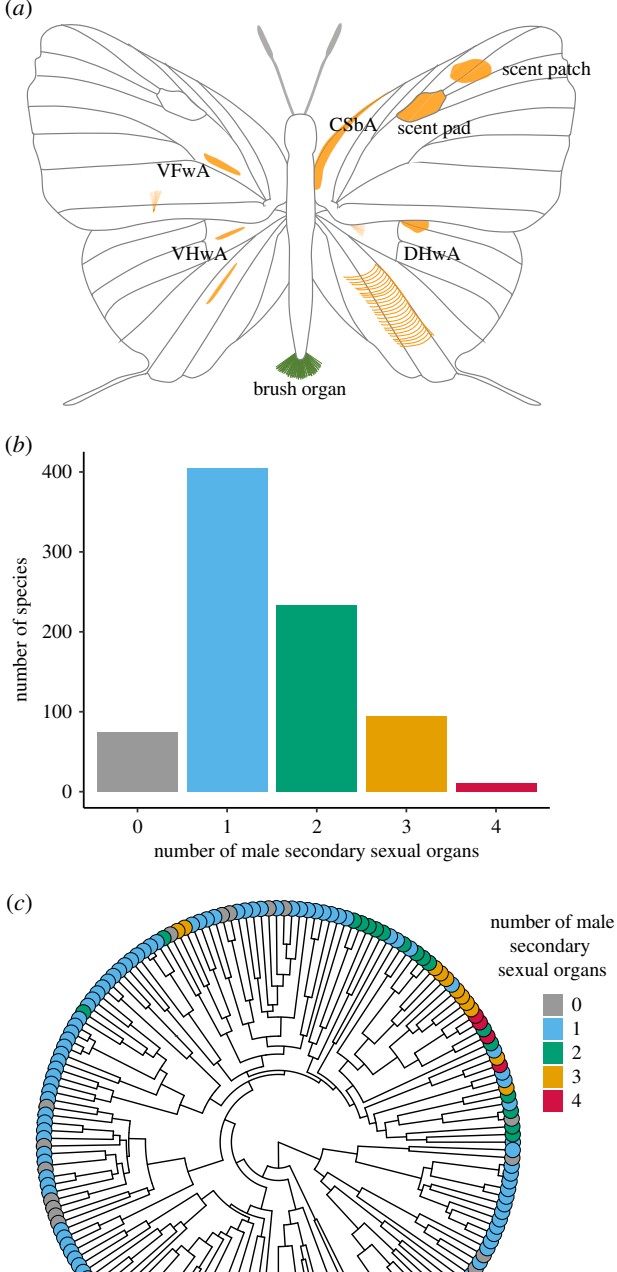

**Figure 2.** Male secondary sexual organs recorded from 818 of the 1,096 described species of Eumaeini. (*a*) Schematic representation of variety and positions of male secondary sexual characters in Eumaeini butterflies. Brush organs, scent pads, and scent patches are widespread in Eumaeini; other characters include: androconia located between the costal and subcostal vein on the dorsal forewing (CSbA), androconia located at the dorsal side of the hindwing (DHwA), androconia located at the ventral side of the forewing (VFwA), and androconia located at the ventral side of the hindwing (VHwA). (*b*) Distribution of the number of species in Eumaeini with multiple male secondary sexual characters. (*c*) Mapping of the number of androconial organs in the Eumaeini phylogeny depicted in figure 1. (Online version in colour.)

organs (*N* = 232; figure 2*b*,*c*), followed by species with three (11.60%, *N* = 95; figure 2*b*,*c*), while only a small fraction exhibit four (1.34%, *N* = 11; figure 2*b*,*c*). Species with three and four male secondary sexual traits clustered phylogenetically, mainly in the *Atlides* section (figure 2*c*).

ML tests favoured models incorporating different transition rates between states (ARD) over the ER for brush organ (*p*-value < 0.0001) and scent patch (*p*-value = 0.005) for the ultrametric tree, but not for scent pad (*p*-value = 0.795), which was more labile across the phylogeny. All reconstruction methods found that the ancestor of Eumaeini likely had brush organs (probability > 0.999, figure 3*a*; electronic supplementary material, figures S12 and S15) with secondary independent losses, notably, in the clade containing *Nicolaea* and *Strephonota* (purple in figure 1) as well as the *Salazaria* and *Penaincisalia* lineage (blue in figure 1). Similarly, scent pads were likely present in the Eumaeini common ancestor (*p* = 0.91, figure 3*a*; electronic supplementary material, figures S13 and S15), with convergent losses across the tree, particularly in the *Calycopis* and *Electrostrymon* section and in the *Salazaria* and *Penaincisalia* clade (pale red in figure 1). Scent patches had a probability of about 97.7% of being present at the root of the tree (electronic supplementary material, figure S14).

ML and Bayesian MCMC methods indicated that possession of a brush organ is negatively correlated with also having a scent pad (ML: mean LTR = 12.362, mean *p*-value = 0.016; MCMC: BF = 8.581, ESS > 9000; figure 3*b*,*c*). Contrastingly, only ML analysis lent support to correlated evolution between the scent pad and the scent patch (mean LTR = 13.92, mean *p*-value = 0.009, MCMC: BF = 1.675, ESS > 5000; electronic supplementary material, figure S17a,b). Lastly, possession of a scent patch is not correlated with having a brush organ under both inference frameworks (mean LTR = 3.21, mean *p*-value = 0.544, MCMC: BF = 0.368, ESS > 7000). Since we consistently recovered a significant negative correlation between brush organs and scent pads across ML (99.4% of the trees were significant, figure 3*a*; electronic supplementary material, figure S12) and Bayesian analyses (strong evidence: BF > 5) (figure 3*b*), we focused our analysis on the relationship between brush organs and scent pads.

Having characterized the dependence between these male secondary sexual traits, we tested hypotheses about conditional evolution and the temporal order of trait acquisition. Under ML and MCMC frameworks, our data are best explained by models where gains and losses have different transition probabilities (AICw = 0.9973, posterior probability = 0.80, figure 4; electronic supplementary material, figure S16), suggesting strong asymmetries in gains and losses. Brush organs were rarely gained, while scent pads were more likely to evolve in lineages that had no androconial organs. In addition, both inference frameworks recovered transition rates q24 and q34 close to zero (figure 4; electronic supplementary material, figure S16), indicating that regaining a brush organ or scent pad is strongly negatively influenced by the presence of the other organ: lineages that have already evolved scent pads are unlikely to regain brush organs and *vice versa*. Under ML, transition rates for a dependent model between pads and patches showed a higher likelihood of gaining a scent pad when a scent patch is already present (q24 on electronic supplementary material, figure S18). Nevertheless, once both traits co-occur, the scent patch is lost more readily than the scent pad (q43 on electronic supplementary material, figure S18).

## 4. Discussion

### (a) Phylogeny of Eumaeini and taxonomic implications

Our phylogenetic results support the hypothesis that Eumaeini is monophyletic [1,25,44]. Monophyly of Eumaeini,

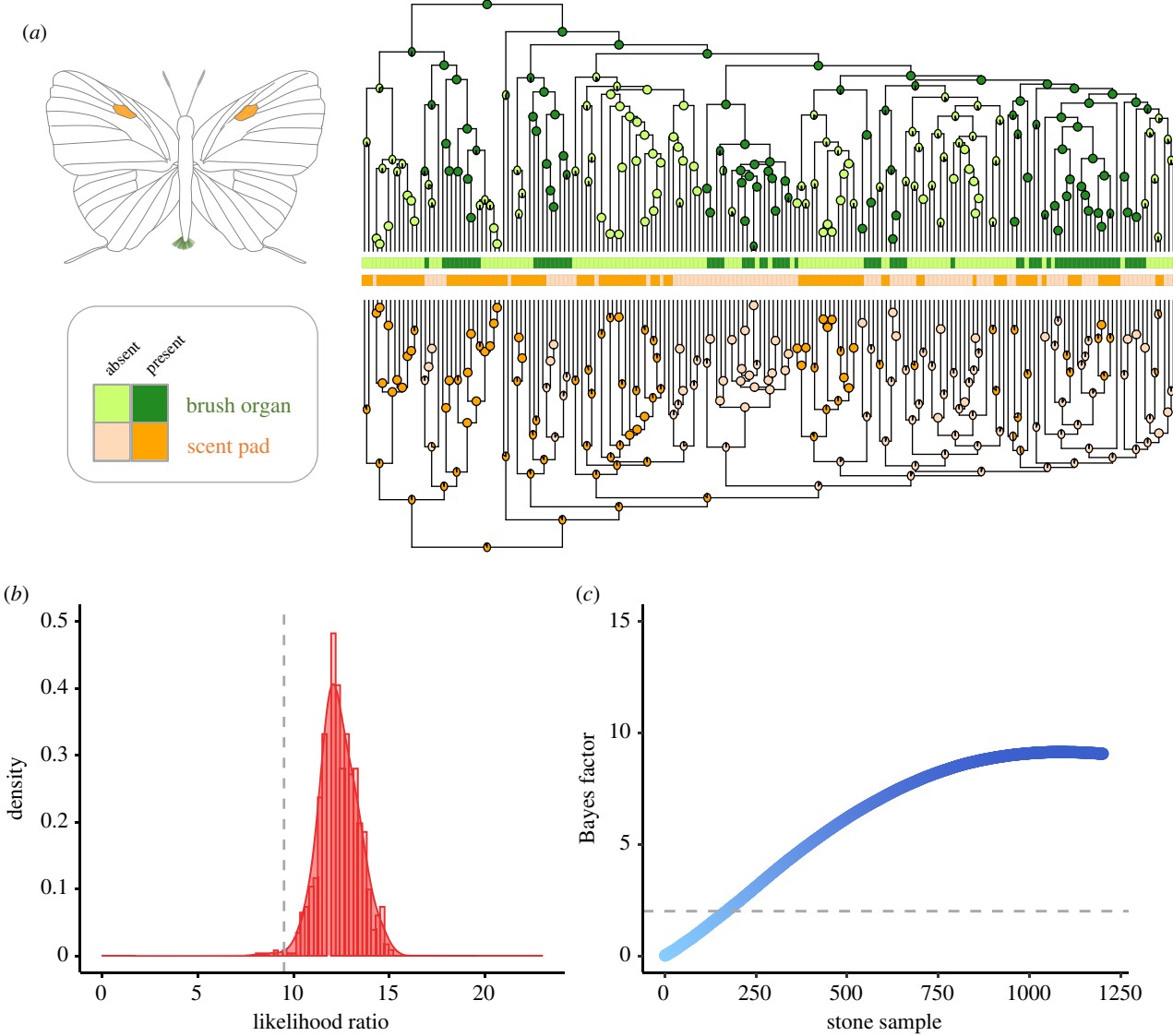

**Figure 3.** Maximum likelihood and Bayesian analyses support a dependent model of evolution between brush organ and scent pad. Males of different species that have one of these organs do not have the other, and *vice versa*. (*a*) Reconstruction of ancestral states for androconial brush organs and scent pads for the 187 species of Eumaeini included in figure 1. Ancestral states were inferred using ML based on 1000 stochastic maps using the ultrametric phylogeny. (*b*) Likelihood ratio tests between independent and dependent models of evolution for the discrete traits 'brush organ' and 'scent pad' for 1000 trees randomly sampled from the distribution of 10 000 bootstraps used to infer the consensus tree. The dashed grey line indicates the significance threshold for which likelihood ratio values >9.5 (*p*-value < 0.05) support significantly correlated evolution; with 99.4% of the trees being significant (right to the dashed line). (*c*) Bayes factors between the independent and dependent model for greater than 1000 MCMC sampling stones with 50 000 iterations. The dashed grey line indicates the significance threshold for which Bayes factor >2 suggests positive and Bayes factor >5 suggests strong evidence in favour of the dependent model (i.e. correlated evolution). (Online version in colour.)

recovered across all ML and coalescent analyses, is also supported by unique morphological characters such as elaborate androconial organs and genitalia that are quite distinct from those of other lycaenid butterflies [45,46]. A striking genitalic trait is that the valvae of many Eumaeini species are inserted into the female during copulation [45,46]. We found Eumaeini to be composed of two large clades that were recovered as monophyletic across all analyses. One is largely a combination of the *Callophrys*, *Erora* and *Satyrium* sections and includes all Palaearctic Eumaeini species. Since our biogeographic analysis indicates that Eumaeini originated in the Neotropics, long-distance dispersal seems likely to have played a central role in the evolution of this group.

The second clade comprises the majority of Eumaeini diversity and is restricted to the New World. Relationships among major lineages in this group showed differences between datasets and analyses (electronic supplementary material, figures S1–S3), likely due to rapid diversification resulting in

high levels of gene tree discordance. Although lower level relationships between species and genera are statistically well supported overall, we recovered short internodes and low bootstrap values for several of the higher level relationships, suggesting insufficient sampling, rampant ILS or introgression. Nevertheless, the topology of the backbone tree is remarkably consistent with a phylogeny using whole genomes that included exemplars from 25 Eumaeini genera [44]. Thus, we suggest that the topological incongruencies we found across analyses are partly caused by ILS due to the explosive diversification of the major Eumaeini clades or introgression between the many closely related species that are sympatric.

Geography appears to have played a central role in the deepest divergences of groups within the Eumaeini. Diversification of all major Eumaeini lineages appears to have co-occurred in the Neotropics during a narrow window between 20 and 25 Ma in the early Miocene. Thus, the geomorphological history and landscape changes across Central and

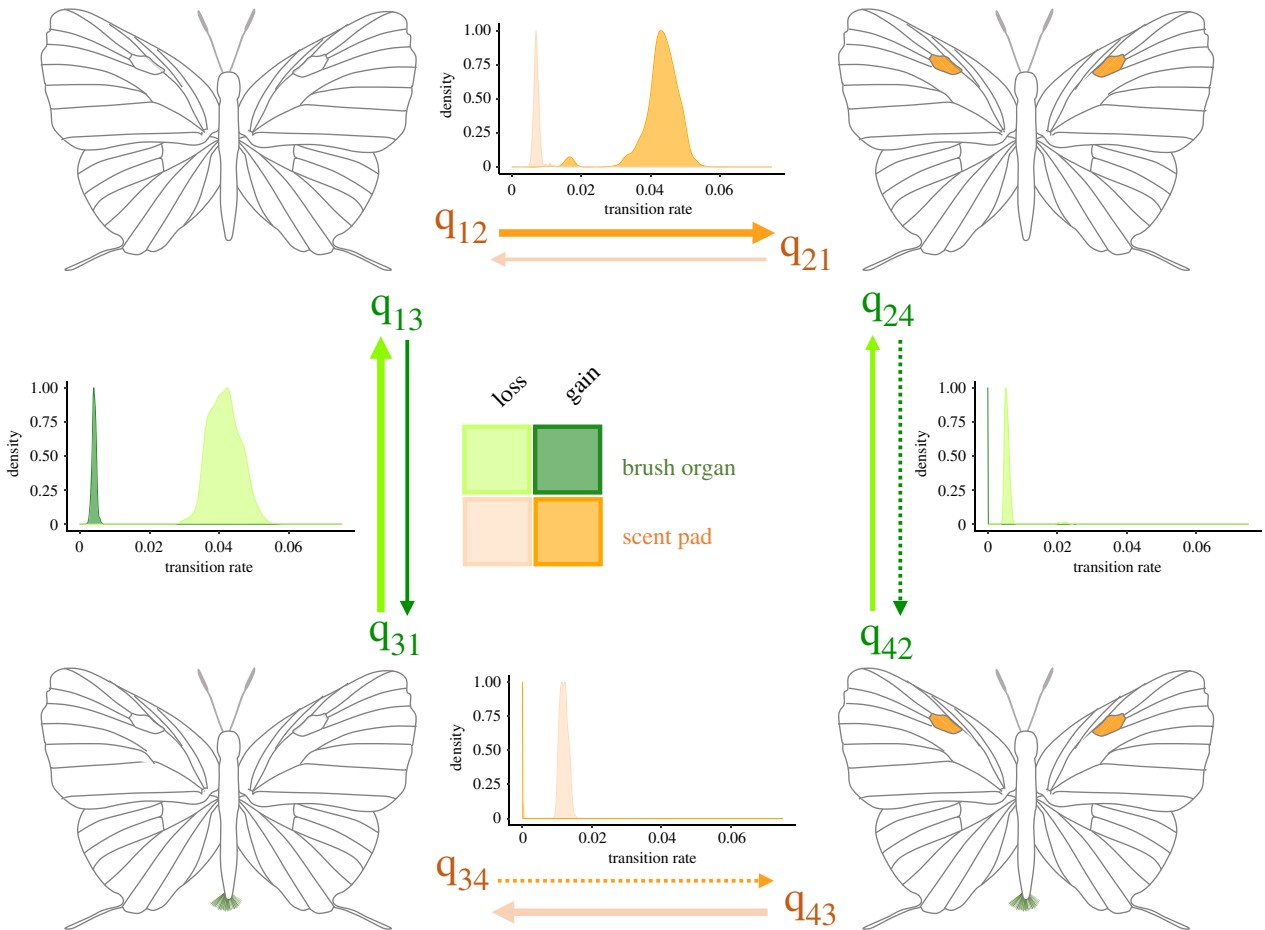

**Figure 4.** Transition rate matrix for the dependent model of evolution between brush organs and scent pads, suggesting that gains are unlikely to occur when males of a species already possess one or the other type of androconial organ. Density plots depict the transition rates for discrete correlated traits inferred using Pagel's method [40] as implemented in Phytools [39] fitted to 1000 randomly sampled trees from the bootstrap distribution used to estimate the support for the consensus tree. Arrow width (approximately) scaled to average rate estimates. Dashed arrows indicate transition rates close to zero based on the ML and Bayesian analyses implementing uncertainty. (Online version in colour.)

South America in the Miocene may offer relevant insights for understanding the evolution of Eumaeini diversity. The tectonic collision between South America and Panama began around 23–25 Ma, ultimately leading to the closure of the Panamanian Isthmus with wide-ranging climatic and biological implications [47]. In South America, mountain orogeny first peaked in this region by early Miocene (approx. 23 Ma), at an age that coincides with the diversification of the first modern montane plant and animal genera [48,49]. Parallel to intensified uplift in the Andes, a large wetland of shallow lakes and swamps developed in Western Amazonia, known as the Pebas Mega-Wetland System, which was fundamental in the evolution of Amazonian landscapes and species composition [48]. The timing of these palaeoenvironmental changes coincide with the divergence of all higher level Eumaeini clades and plausibly exerted a significant influence on the rapid evolution of the tribe. Along with these environmental factors, Eumaeini diversification was likely to have been generated by biotic interactions, as evidenced in part by the widespread distribution of sexually selected innovations such as brush organs and scent pads and patches on male wings.

## (b) Evolution of secondary sexual characteristics within the Eumaeini

A key insight from our study is the evolution of a negative co-dependence across the tribe between two male secondary traits,

scent pads and brush organs. The negative trade-off between these traits is likely to be mediated by functional redundancy or by a greater role that the more conspicuous signal plays in female choice and intraspecific recognition [18,50,51]. Brush organs and scent pads were not regained in species that already had one of these sexual characteristics. Conversely, scent patches seem to facilitate the evolution of the scent pads. Still, once both traits are present, scent patches are lost at a higher frequency, pointing to possible functional redundancy between these two types of wing androconia. These dynamics of gains and losses across Eumaeini underscore the role of selection on sexual traits maintaining at least one of these organs involved in courtship and species recognition.

Both brush organs and scent pads appear to have evolved in the common ancestor of Eumaeini and were subsequently lost several times, and in the case of the brush organ, rarely regained (figure 4c; electronic supplementary material, figure S13). A higher rate of losses of sexually selected male traits is a widespread evolutionary trend and is consistent with Dollo's Law, which states that reversals to former states are rare probabilistic events [20,52]. Therefore, given a character that evolved early in a large clade, a higher frequency of losses should be expected [20]. Nonetheless, this expectation rests on the assumption that the evolution of the traits is random, whereas if sexual selection is the primary responsible for the origin and maintenance of these characteristics, then traits should only be lost when other forces such

as drift or natural selection are strong enough to override sexual selection [20].

Male secondary sexual organs can have high costs in terms of natural selection through predation risk, signal transmission and nutrient availability [18,20,50,51]. Due to these associated costs, natural selection may outweigh sexual selection, constraining the evolution of male sexual characters and leading to evolutionary trade-offs [20,50]. We observed that the majority of Eumaeini (about 60%) retained only one secondary sexual organ or lost them completely. It seems likely that natural selection constrains Eumaeini males from maintaining multiple attractive cues through favouring losses and/or hindering regains of redundant sexual characters. Negative correlations between male sexual traits have also been found in horn beetles, dobsonflies, finches, toucans and barbets, and Old World monkeys and apes [18,53–55]. For example, in horned beetles, an increase in the investment into head horns across 11 species coincided with a decrease in aedeagus investment [55]. Analogously, in dobsonflies, when male weapons are large, nuptial gifts are small and *vice versa* [56]. Finally, in finches, song complexity is negatively related to the elaboration of plumage ornamentation, indicating a trade-off in costs or in the information content of these traits [18].

Losses of secondary male traits are often associated with habitat changes, the presence of male territoriality or parental care, and/or the presence of sympatric species with similar traits [20,50]. In particular, male traits can be lost if they are involved in intraspecific recognition, and strong selection against hybridization is relaxed when other barriers preventing gene flow are at play. In Eumaeini, Robbins *et al.* [14,15] showed that for the genera *Arcas* and *Thereus,* evolutionary losses of the scent pad occurred in species that were geographically isolated from their closest relatives. Similarly, Martins *et al.* [13] found that the scent pad and brush organs were lost recurrently in clades of the *Atlides* section allopatrically distributed with their sisters. By contrast, these authors found regains of the scent pad in lineages that were sympatric with their sister species [13]. Remarkably, the gains of the scent pad found by Martins *et al.* [13] in *Atlides* lend support to the single scenario, according to our transition matrix, whereby gaining a scent pad is more probable than losing it, which is when the ancestor does not also have brush organs. This highlights the potential importance of male secondary sexual organs as isolating mechanisms in speciation between recently diverged, incipient species. Additionally, multiple male traits as a mechanism aiding reproductive isolation might become less important if closely related species discriminate between the chemical signals produced by these androconia. For instance, *Heliconius* butterflies do not vary significantly in their number of androconial organs; however, there is striking between-species variation in chemical bouquets [57,58]. These chemical blends drive female preference for conspecific males in sympatric, co-mimetic species, and seem to evolve rapidly with pheromone gains and losses occurring frequently across the *Heliconius* phylogeny [58–60]. Hence, rapid divergent chemical evolution might render the loss of multiple secondary sexual characters more likely. Currently, the chemical ecology of individual species of Eumaeini remains largely unknown, but the macroevolutionary patterns shown here suggest that this would be a productive avenue for future research.

Similar to the scent pad, the best fit model for the evolution of the scent patch predicted non-zero gain probabilities, denoting that these wing androconia are more labile than the brush organ. For instance, Prakash & Monteiro [61] found that scent patches were gained multiple times during the evolution of *Bicyclus*. They further showed that the molecular basis of scent patch formation in *Bicyclus* and *Orsotriaena* butterflies is determined by the spatial regulation of the gene *doublesex*, similar to the regulation of this gene in sex-comb initiation in *Drosophila* [61]. It is plausible that this developmental mechanism for male-specific secondary organs is conserved not only across Lepidoptera, but across insects more broadly, and consequently plays a critical role in Eumaeini alar androconial development. The scent pad is unique to Eumaeini, and in addition to scent cells, it contains an elaborate network of haemolymph channels subtended along a primary wing vein by a 'wing heart' that beats endogenously [16]. Although the scent pad is more complex, it might share a developmental mechanism with scent patches given their locations and their tendency to appear together in different regions of the wing. This is preliminarily supported by our transition rates results indicating that gaining a scent pad is more likely when a scent patch has already evolved. Prakash & Monteiro [61] suggest that diversification in the number and location of the patches in the *Bicyclus* radiation occurred via gain and loss of new domains of *doublesex* expression in the wing, and it seems possible that similar genetic machinery governs the multiple male pheromone-producing organs in the wings of Eumaeini males (figure 2*a*).

The most comprehensive existing survey of androconial organs across higher level groups of butterflies is for Riodinidae [62]. Interestingly, Riodinidae parallels Eumaeini in species richness and morphological variation in male traits. Despite these similarities, the males of only 25% of riodinid species have androconial organs [62], whereas this study showed that males of 91% of Eumaeini species surveyed here show these traits. Therefore, species of Eumaeini exhibit among the most striking examples of selection for secondary sexual organs within butterflies [1,13,15], and it seems plausible that an interaction between sexual selection and selection for species recognition may have contributed to the establishment of early reproductive barriers, potentially contributing to their extraordinary radiation. A robust phylogeny of the Lycaenidae, along with a more comprehensive review of androconial organs across the family, will facilitate systematic testing of associations between the extent of sexual selection and diversification in the Eumaeini.

**Data accessibility.** Research data associated with this study are available from the Dryad Digital Repository: https://doi.org/10.5061/dryad.tqjq2bvz9 [63], and NCBI Bioproject PRJNA714105, Biosamples: SAMN18315678–SAMN18315893 and SAMN18322226–SAMN18322244.

**Authors' contributions.** All authors gave final approval for publication and agreed to be held accountable for the work performed therein.

**Competing interests.** We declare we have no competing interests.

**Funding.** This study was supported by Instituto Chico Mendes de Conservação da Biodiversidade (grant no. 11990-1), David Rockefeller Center for Latin American Studies (DRCLAS), Society for Systematic Biology (SSB- Mini-PEET), DRCLAS and Lemann Foundation, "Ramón y Cajal" programme of the Spanish Ministry of Science and Innovation (grant no. RYC2018-025335-I), Putnam Expeditionary Fund, Museum of Comparative Zoology, Department of Organismic and Evolutionary Biology (OEB) at Harvard University, Conselho Nacional de Desenvolvimento Científico e Tecnológico (CNPq) (grant

nos. 200814/2015-0 and 304273/2014-7), Instituto Chico Mendes de Conservação da Biodiversidade (ICMBio) (Collecting license (11990-1) and Collecting license (20395-1)) and NSF (grant nos. 1906333, DEB-0447244, DEB-1541500, DEB-1541557 and DEB-1541560).

Acknowledgements. We thank Michael F. Braby, Dana Campbell, James Coleman, Mark Cornwall, Alexandre Danchenko, Kelvyn Dunn, Rod Eastwood, Alan Heath, Nikolai Kandul, Norbert G. Kondla, N. Mega, Carlos Pena, Jon Sanders, Art Shapiro, Man Wah Tan, Carlos Tello, Andrew D. Warren, Doug W. Yu and Roger Vila for collecting specimens used in this research. Our colleagues Mark Cornwall, Rod Eastwood, Charles Marshall, Manus Patten, Swee Peck Quek, Santiago Ramírez, Andrei Sourakov, Benjamin Goldman-Huertas, Sam Church, Neil Rosser and Mark Wright provided help in numerous ways and/or made helpful comments on the manuscript. We thank the webpage: www.butterfliesofamerica.com for the butterfly pictures used in Figure 1. Published by a grant from the MCZ Wetmore Colles fund.

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
