## [Peer Review File · Proceedings of the Royal Society B: Biological Sciences]

Review History

RSPB-2020-2512.R0 (Original submission)

Review form: Reviewer 1

Recommendation

Accept with minor revision (please list in comments)

Scientific importance: Is the manuscript an original and important contribution to its field?

Excellent

General interest: Is the paper of sufficient general interest?

Good

Quality of the paper: Is the overall quality of the paper suitable?

Good

Is the length of the paper justified?

Yes

Should the paper be seen by a specialist statistical reviewer?

No

Do you have any concerns about statistical analyses in this paper? If so, please specify them explicitly in your report.

Yes

It is a condition of publication that authors make their supporting data, code and materials available - either as supplementary material or hosted in an external repository. Please rate, if applicable, the supporting data on the following criteria.

Is it accessible?

Yes

Is it clear?

Yes

Is it adequate?

No

Do you have any ethical concerns with this paper?

No

Comments to the Author

The authors reconstructed the phylogeny of one large butterfly group based on hundreds of loci. They also used the phylogeny to infer the evolution of male secondary sexual traits in this group. This manuscript is well written and the inferences of evolutionary relationship among multiple secondary sexual traits are sound. But I have some comments on the analytic details and discussion (see below). I also have one suggestion that even though the focus of this study is in male sexual traits, females' traits should be somehow genetically or developmentally related to the male ones and subject to relevant natural selection, and the potential association may be worthy for further discussion. Overall, I think that this study may significantly contribute to our understanding of secondary sexual trait evolution and I enjoy reading the manuscript.

Line 187-188: What were the criteria, based on which you determined the ten likelihood searches sufficient to ensure convergence among the runs?

Line 197-198: Didn't the complete concatenated matrix contain 14 loci, 191 spp.?

Line 215-218: This sentence is not clear enough. What was your reason to choose F84 substitution model? Was it because the data set small or large?

Line 279-292: Did you use any approach to ensure adequate mixing and/or high effective sample size (ESS) for posterior probability distributions? If yes, please add the info. If not, you should at least clarify how you assessed convergence among the MCMC runs.

Line 319-320: Does this result mean that the 12 genera need to be renamed or the phylogeny might be biased by sampling gaps? It may be worthy of further discussions.

Line 426-430: I do not find such information in Fig. 4a and 4b.

Line 504: Do you mean "many of the gene trees"?

Line 541-547: You argue that brush organs and scant pads were not regained in species that already had one of them due to functional redundancy. However, your results showed that scant

patches facilitated the gain of scant pads. Does it mean that the functional redundancy level between scant patches and scant pads was lower than that between brush organs and scant pads?

Line 561: reversals of what?

Line 665-667: Do you mean that the number and complexity of male secondary sexual traits “are correlated with” the intensity of sexual selection?

Line 716: For Research, I suggest that the sequence data should be deposited to the NCBI Genbank so that they will have better chances to be used in the future.

Review form: Reviewer 2

Recommendation

Accept as is

Scientific importance: Is the manuscript an original and important contribution to its field?

Good

General interest: Is the paper of sufficient general interest?

Good

Quality of the paper: Is the overall quality of the paper suitable?

Excellent

Is the length of the paper justified?

Yes

Should the paper be seen by a specialist statistical reviewer?

No

Do you have any concerns about statistical analyses in this paper? If so, please specify them explicitly in your report.

No

It is a condition of publication that authors make their supporting data, code and materials available - either as supplementary material or hosted in an external repository. Please rate, if applicable, the supporting data on the following criteria.

Is it accessible?

Yes

Is it clear?

Yes

Is it adequate?

Yes

Do you have any ethical concerns with this paper?

No

Comments to the Author

This is an exciting manuscript reporting a higher level phylogenetic hypothesis for a species rich group of butterflies, and an analysis of how the evolution of male secondary sexual characters may have affected their diversification. I find no fault in the analyses or the interpretation of the results. A very nice study!

Review form: Reviewer 3**Recommendation**

Accept with minor revision (please list in comments)

Scientific importance: Is the manuscript an original and important contribution to its field?

Good

General interest: Is the paper of sufficient general interest?

Good

Quality of the paper: Is the overall quality of the paper suitable?

Good

Is the length of the paper justified?

Yes

Should the paper be seen by a specialist statistical reviewer?

No

Do you have any concerns about statistical analyses in this paper? If so, please specify them explicitly in your report.

No

It is a condition of publication that authors make their supporting data, code and materials available - either as supplementary material or hosted in an external repository. Please rate, if applicable, the supporting data on the following criteria.

Is it accessible?

Yes

Is it clear?

Yes

Is it adequate?

No

Do you have any ethical concerns with this paper?

No

Comments to the Author

This paper resolved to explore the evolution and maintenance of male secondary sexual traits across the Lepidopteran tribe, Eumaeini. They used an AHE approach for a subset of 78 species (including 4 outgroups) to generate a phylogenomic data based on ~160k bp per species (378 loci) to generate a backbone as a constraint for full ML analyses that included additional data for 113 species based on ~11k bp per species (13 loci). Although Eumaeini is monophyletic, the authors point out several groups are non-monophyletic, but the major clades corresponded to main

biogeographic events. In addition, interestingly, the evolution of several secondary male traits are co-dependent with strong asymmetries in gains and losses.

The scope of the paper is impressive, with substantial taxa and loci representation across ~90% of the genera of the Eumaeini. Analyses selected by the authors for reconstruction of the phylogeny and phenotypic evolution seem appropriate. The phylogeny presented by the authors is also relatively well resolved with good support. Overall, I think the manuscript is well-written and the narrative is clear. However, I do have several specific comments that I have detailed below:

Specific comments

L78-80: How are these results consistent with the trade-off between natural and sexual selection?

I suggest removing this sentence because I do not see the connection in this paragraph. I.e. where are the 'natural selection' traits that are trading off with the sexual traits? Instead I think the authors can just proceed with "Our results illustrate...".

L125: I think the introduction is a little sparse to be honest. I think authors can expand a bit more on the trade-offs component since this plays a big role in their overall narrative.

L144-145: I think this section could be more informative. How were the samples collected? What is the provenance and condition of samples, i.e. all samples from museum collections, in ethanol or snap frozen? All adult tissue or larvae from cultures? It will be useful to include a supplementary file detailing specifics of each sample used in the genomic study.

L 155: Were whole samples (including wings) used for the extraction or just specific body parts?

L 156: gbiosciences.com should be replaced with the manufacturer name and country

L 157: What was the sequencing platform used?

L 198 and 203: Is the same dataset being referred to here?

L 233: "We gathered additional data on morphological traits from recently published studies".

Purely male sexual traits? How about other traits including overall body size or any female traits?

L 245: Missing (VHwA)

L 246: From each of the 819 species?

L238-245: The section on androconial organs seem rather abrupt since nothing was mentioned in the introduction. I think the inclusion of these traits need to be justified and explained in the introduction.

L299: I think the authors need to at least include a line or two regarding the comparison between the constraint (i.e with the phylogenomic backbone) versus non-constraint approach since they mention it in the methods (L199). I did not find it referenced in the supplementary document either.

L288-292: Not convinced of why authors use Pearson momentum correlation test. Why exclude phylogenetic relatedness?

L 363: Caption missing (VHwA)

L 499-502: the authors mentioned that introgression could also be a reason for the lack of resolution in the higher level phylogeny in L 317, might be good to discuss it here as well.

L 541: "...negative tradeoff between these traits is likely to be mediated by functional redundancy..." This is a rather broad generalization. At the molecular level, the functions of these organs are more likely to be species specific, aiding in species recognition. The authors touch upon this later (L 678-680: "...it seems plausible that an interaction between sexual selection and selection for species recognition may have contributed to the establishment of early reproductive barriers..."). Selection acting on peptides and proteins that these organs secrete likely play an important role in the usefulness of these organs and could thereby ultimately decide their gain/loss/retention in a species. These need to be considered in the discussion.

Language comments

Overall the language structure fluctuates between passive and active voice. I urge the authors to stick to one.

L69: "We examined specimens from 819 of the 1,096 described species (75%) and documented that male secondary sexual traits are present in 91% of the species in our dataset." Not accurate to say 91% of known species if only examined a subset.

L71: Quotations not necessary for sexual traits.

L72: "... were likely present in..."

Decision letter (RSPB-2020-2512.R0)

20-Jan-2021

Dear Dr Valencia-Montoya:

Thank you for the submission of this manuscript, which has now been peer reviewed and the reviews have been assessed by an Associate Editor. I apologise that this has taken longer than we would usually hope, due to the combination of the Christmas break and the pandemic slowing the world down. The reviewers' comments (not including confidential comments to the Editor) and the comments from the Associate Editor are included at the end of this email for your reference. We are all agreed that this is a potentially really valuable and interesting manuscript. However the reviewers and the Editors have raised some concerns with the manuscript in its current form, which we would therefore like to invite you to address.

Research ethics:

Use of animals and field studies:

It is a condition of publication that you make available the data and research materials supporting the results in the article. Please see our Data Sharing Policies (<https://royalsociety.org/journals/authors/author-guidelines/#data>). Datasets should be deposited in an appropriate publicly available repository and details of the associated accession number, link or DOI to the datasets must be included in the Data Accessibility section of the article (<https://royalsociety.org/journals/ethics-policies/data-sharing-mining/>). Reference(s) to datasets should also be included in the reference list of the article with DOIs (where available).

Please submit a copy of your revised paper within three weeks. If we do not hear from you within this time your manuscript will be rejected. If you are unable to meet this deadline please let us know as soon as possible, as we may be able to grant a short extension.

Finally, I hope you and co-authors have been well in these difficult times; wishing you all the best for the New Year, and hoping for a calm and healthy 2021.

Best wishes,
Professor Loeske Kruuk
<mailto:proceedingsb@royalsociety.org>

Associate Editor
Board Member: 1
Comments to Author:

This manuscript used sequence information from 378 loci to reconstruct the phylogeny for a species rich butterfly family, Eumaeini. Based this phylogeny, authors studied trait evolution of secondary sexual characteristics involved in pheromone production and dissemination in

Eumaeini. They concluded that evolution of male secondary sexual signaling in these butterflies involving tradeoffs between natural and sexual selection. As pointed out by all three reviewers, the data and analysis of this research are sound and the manuscript is well written. Their results provide good insights on how different selection forces worked hand-in-hand to shape evolution of male secondary sexual signaling system in this group of butterflies. It should be interesting to a wide range of potential readers. However, as pointed out by the reviewer 1 and 3, some details have to be further clarified before I can recommend accepting this manuscript for publication in PRSB. I urge authors to address all the points raised by the reviewer when they revise their manuscript accordingly.

Reviewer(s)' Comments to Author:

Referee: 1

Comments to the Author(s)

The authors reconstructed the phylogeny of one large butterfly group based on hundreds of loci. They also used the phylogeny to infer the evolution of male secondary sexual traits in this group. This manuscript is well written and the inferences of evolutionary relationship among multiple secondary sexual traits are sound. But I have some comments on the analytic details and discussion (see below). I also have one suggestion that even though the focus of this study is in male sexual traits, females' traits should be somehow genetically or developmentally related to the male ones and subject to relevant natural selection, and the potential association may be worthy for further discussion. Overall, I think that this study may significantly contribute to our understanding of secondary sexual trait evolution and I enjoy reading the manuscript.

Line 187-188: What were the criteria, based on which you determined the ten likelihood searches sufficient to ensure convergence among the runs?

Line 197-198: Didn't the complete concatenated matrix contain 14 loci, 191 spp.?

Line 215-218: This sentence is not clear enough. What was your reason to choose F84 substitution model? Was it because the data set small or large?

Line 279-292: Did you use any approach to ensure adequate mixing and/or high effective sample size (ESS) for posterior probability distributions? If yes, please add the info. If not, you should at least clarify how you assessed convergence among the MCMC runs.

Line 319-320: Does this result mean that the 12 genera need to be renamed or the phylogeny might be biased by sampling gaps? It may be worthy of further discussions.

Line 426-430: I do not find such information in Fig. 4a and 4b.

Line 504: Do you mean "many of the gene trees"?

Line 541-547: You argue that brush organs and scant pads were not regained in species that already had one of them due to functional redundancy. However, your results showed that scant patches facilitated the gain of scant pads. Does it mean that the functional redundancy level between scant patches and scant pads was lower than that between brush organs and scant pads?

Line 561: reversals of what?

Line 665-667: Do you mean that the number and complexity of male secondary sexual traits "are correlated with" the intensity of sexual selection?

Line 716: For Research, I suggest that the sequence data should be deposited to the NCBI Genbank so that they will have better chances to be used in the future.

Referee: 2

Comments to the Author(s)

This is an exciting manuscript reporting a higher level phylogenetic hypothesis for a species rich group of butterflies, and an analysis of how the evolution of male secondary sexual characters may have affected their diversification. I find no fault in the analyses or the interpretation of the results. A very nice study!

Referee: 3

Comments to the Author(s)

This paper resolved to explore the evolution and maintenance of male secondary sexual traits across the Lepidopteran tribe, Eumaeini. They used an AHE approach for a subset of 78 species (including 4 outgroups) to generate a phylogenomic data based on ~160k bp per species (378 loci) to generate a backbone as a constraint for full ML analyses that included additional data for 113 species based on ~11k bp per species (13 loci). Although Eumaeini is monophyletic, the authors point out several groups are non-monophyletic, but the major clades corresponded to main biogeographic events. In addition, interestingly, the evolution of several secondary male traits are co-dependent with strong asymmetries in gains and losses.

The scope of the paper is impressive, with substantial taxa and loci representation across ~90% of the genera of the Eumaeini. Analyses selected by the authors for reconstruction of the phylogeny and phenotypic evolution seem appropriate. The phylogeny presented by the authors is also relatively well resolved with good support. Overall, I think the manuscript is well-written and the narrative is clear. However, I do have several specific comments that I have detailed below:

Specific comments

L78-80: How are these results consistent with the trade-off between natural and sexual selection? I suggest removing this sentence because I do not see the connection in this paragraph. I.e. where are the 'natural selection' traits that are trading off with the sexual traits? Instead I think the authors can just proceed with "Our results illustrate...".

L125: I think the introduction is a little sparse to be honest. I think authors can expand a bit more on the trade-offs component since this plays a big role in their overall narrative.

L144-145: I think this section could be more informative. How were the samples collected? What is the provenance and condition of samples, i.e. all samples from museum collections, in ethanol or snap frozen? All adult tissue or larvae from cultures? It will be useful to include a supplementary file detailing specifics of each sample used in the genomic study.

L 155: Were whole samples (including wings) used for the extraction or just specific body parts?

L 156: gbiosciences.com should be replaced with the manufacturer name and country

L 157: What was the sequencing platform used?

L 198 and 203: Is the same dataset being referred to here?

L 233: "We gathered additional data on morphological traits from recently published studies". Purely male sexual traits? How about other traits including overall body size or any female traits?

L 245: Missing (VHwA)

L 246: From each of the 819 species?

L238-245: The section on androconial organs seem rather abrupt since nothing was mentioned in the introduction. I think the inclusion of these traits need to be justified and explained in the introduction.

L299: I think the authors need to at least include a line or two regarding the comparison between the constraint (i.e with the phylogenomic backbone) versus non-constraint approach since they mention it in the methods (L199). I did not find it referenced in the supplementary document either.

L288-292: Not convinced of why authors use Pearson momentum correlation test. Why exclude phylogenetic relatedness?

L 363: Caption missing (VHwA)

L 499-502: the authors mentioned that introgression could also be a reason for the lack of resolution in the higher level phylogeny in L 317, might be good to discuss it here as well.

L 541: "...negative tradeoff between these traits is likely to be mediated by functional redundancy..." This is a rather broad generalization. At the molecular level, the functions of these organs are more likely to be species specific, aiding in species recognition. The authors touch upon this later (L 678-680: "...it seems plausible that an interaction between sexual selection and selection for species recognition may have contributed to the establishment of early reproductive barriers..."). Selection acting on peptides and proteins that these organs secrete likely play an important role in the usefulness of these organs and could thereby ultimately decide their gain/loss/retention in a species. These need to be considered in the discussion.

Language comments

Overall the language structure fluctuates between passive and active voice. I urge the authors to stick to one.

L69: "We examined specimens from 819 of the 1,096 described species (75%) and documented that male secondary sexual traits are present in 91% of the species in our dataset." Not accurate to say 91% of known species if only examined a subset.

L71: Quotations not necessary for sexual traits.

L72: "... were likely present in..."

Author's Response to Decision Letter for (RSPB-2020-2512.R0)

See Appendix A.

RSPB-2020-2512.R1 (Revision)

Review form: Reviewer 1 (Chih-Ming Hung)

Recommendation

Accept as is

Scientific importance: Is the manuscript an original and important contribution to its field?

Excellent

General interest: Is the paper of sufficient general interest?

Good

Quality of the paper: Is the overall quality of the paper suitable?

Excellent

Is the length of the paper justified?

Yes

Should the paper be seen by a specialist statistical reviewer?

No

Do you have any concerns about statistical analyses in this paper? If so, please specify them explicitly in your report.

No

It is a condition of publication that authors make their supporting data, code and materials available - either as supplementary material or hosted in an external repository. Please rate, if applicable, the supporting data on the following criteria.

Is it accessible?

Yes

Is it clear?

Yes

Is it adequate?

Yes

Do you have any ethical concerns with this paper?

No

Comments to the Author

The authors have carefully answered my comments and made necessary changes in the manuscript. I am satisfied by their revision and have no further comments.

Decision letter (RSPB-2020-2512.R1)

19-Apr-2021

Dear Dr Valencia-Montoya

I am pleased to inform you that your manuscript entitled "Evolutionary tradeoffs between male secondary sexual traits revealed by a phylogeny of the hyperdiverse tribe Eumaeini (Lepidoptera: Lycaenidae)" has been accepted for publication in Proceedings B.

Data Accessibility section

Open Access

Paper charges

Sincerely,

Professor Loeske Kruuk

Associate Editor:

Board Member: 1

Comments to Author:

(There are no comments.)

Board Member: 2

Comments to Author:

(There are no comments.)

HARVARD UNIVERSITY
DEPARTMENT OF ORGANISMIC AND EVOLUTIONARY BIOLOGY
26 OXFORD STREET
Cambridge, Massachusetts 02138 USA

25 February 2021

Professor Loeske Kruuk
Proceedings of the Royal Society B
The Royal Society
6-9 Carlton House Terrace,
London SW1Y 5AG

RE: Evolutionary tradeoffs between male secondary sexual traits revealed by a phylogeny of the hyperdiverse tribe Eumacini (Lepidoptera: Lycaenidae)

Dear Professor Kruuk and Associate Editor 1,

Thank you and the reviewers for your comprehensive and constructive feedback on our manuscript. We are glad that you agree that the manuscript addresses an interesting question, that our data and analyses are sound and that the potential impact of our findings is high and interesting to a wide range of potential readers. We have now thoroughly revised the manuscript to address the issues raised by the reviewers. In particular, we have improved the introduction and the discussion, and we have added substantial details about our taxon sampling, sequencing methods, and convergence statistics for our multiple Bayesian analyses. Because of stringent length restrictions, we have also implemented some of the changes suggested by the reviewers in the Supplementary Material.

We believe the comments and criticisms have greatly improved the manuscript, and hope that we have addressed all the reviewers' comments.

Please find our point-by-point response to the reviewers below.
We look forward to your response.

Yours sincerely, on behalf of the author team,

Naomi E. Pierce

Point-by-point response to reviewers' comments

Note that line numbers correspond to manuscript with tracked changes included at the end of this response.

Referee 1

The authors reconstructed the phylogeny of one large butterfly group based on hundreds of loci. They also used the phylogeny to infer the evolution of male secondary sexual traits in this group. This manuscript is well written and the inferences of evolutionary relationship among multiple secondary sexual traits are sound. But I have some comments on the analytic details and discussion (see below). I also have one suggestion that even though the focus of this study is in male sexual traits, females' traits should be somehow genetically or developmentally related to the male ones and subject to relevant natural selection, and the potential association may be worthy for further discussion. Overall, I think that this study may significantly contribute to our understanding of secondary sexual trait evolution and I enjoy reading the manuscript.

1.1. Line 187-188: *What were the criteria, based on which you determined the ten likelihood searches sufficient to ensure convergence among the runs?*

We followed IQ-TREE 'best practices' and used the variance among tree log-likelihoods and tree topology as the criteria to determine whether we have found the optimal tree. We found minimal log-likelihood variance among IQ-TREE runs, as well as negligible topological discordance between trees, indicating that our phylogenomic data set contains sufficient phylogenetic signal. The graph below shows the pairwise Robinson-Foulds distance between the trees resulting from 10 runs, showing that only three trees (e.g., 3, 4, and 5) differed in the location of two taxa, which is the minimal distance possible between trees.

We have now modified the text to clarify the criteria we used to determine that 10 runs were sufficient as follows:

“We used the variance among tree log-likelihoods and differences in tree topology as indicated by Robinson-Foulds distances as the criteria to identify the optimal tree.” (L181-182)

1.2. Line 197-198: *Didn't the complete concatenated matrix contain 14 loci, 191 spp.?*

We thank the reviewer for spotting this mistake. The concatenated matrix indeed had 14 loci, and we have corrected it accordingly. (L197)

1.3. Line 215-218: *This sentence is not clear enough. What was your reason to choose F84 substitution model? Was it because the data set small or large?*

We thank the reviewer for catching this ambiguity. Following (Espeland et al. 2018), we selected the F84 substitution model because of the large size of the dataset and because we wanted to carry out divergence time analyses in a computationally feasible manner. We have now noted that the dataset was “large” in the text (L212). Although this model differentiates between transitions and transversions, allows for unequal base composition, and even has a more complicated instantaneous rate matrix than the HKY85, it is nevertheless mathematically more tractable and performs better on datasets with a large number of species (McGuire et al. 2001).

1.4. Line 279-292: *Did you use any approach to ensure adequate mixing and/or high effective sample size (ESS) for posterior probability distributions? If yes, please add the info. If not, you should at least clarify how you assessed convergence among the MCMC runs.*

For analyses of trait evolution in BayesTraits v.3.0.0 (Pagel, 1994), we calculated ESS (effective sample size) as the convergence diagnostic for the MCMC runs using the R package “coda” (Plummer *et al.* 2006). For all runs, we recovered high effective sample sizes (ESS > 5000), thus indicating that the continuous chains were well-mixing. We have now added this to the manuscript in lines L290-291 and L407-L411.

Additionally, following the MCMCtree manual for estimation of divergence times (dos Reis et al. 2019), we assessed convergence among MCMC runs plotting the posterior mean times from the best two runs, checking that the two sets of posterior mean times fell on a straight line with intercept 0 and slope 1, as shown below:

1.5. Line 319-320: *Does this result mean that the 12 genera need to be renamed or the phylogeny might be biased by sampling gaps? It may be worthy of further discussions.*

We thank the reviewer for this question as it helped us to identify that there are 13 non-monophyletic genera rather than the 12, we originally stated. This is because some species were recently transferred to other genera or synonymized. For instance, *Theritas* forms two monophyletic clusters, one of which corresponds to the recently identified genus *Denivia* (Martins et al. 2019). We believe that sampling gaps at the generic level have not biased the recovery of these genera as non-monophyletic, as they include species that are notoriously difficult to place based on morphology. Nonetheless, since four of these genera included more than two species: *Chalybs*, *Electrostrymon*, *Michaelus*, and *Ziegleria* (they only have three included species) our phylogeny is not sufficient to provide guidance about renaming these non-monophyletic genera beyond simply pointing out the discrepancy between molecular and morphology placements, which needs to be addressed in the future. We have now included a list of the 13 genera that were not recovered as monophyletic, as well as further discussion in the Supplementary Material.

1.6. Line 426-430: *I do not find such information in Fig. 4a and 4b.*

We thank the reviewer from catching this mistake, we actually meant Figure 3a and 3b. We have now change this accordingly (L420).

1.7. Line 504: *Do you mean “many of the gene trees”?*

We thank the reviewer for spotting this. We have now added “many of the” to the text, and we have decided to move this section to Supplementary discussion due to length restrictions.

1.8 Line 541-547: *You argue that brush organs and scent pads were not regained in species that already had one of them due to functional redundancy. However, your results showed that scent patches facilitated the gain of scent pads. Does it mean that the functional redundancy level between scent patches and scent pads was lower than that between brush organs and scent pads?*

Thank you for raising this interesting question. We think that the key to this interpretation is the order of trait acquisition shown by the transition matrix of the model of correlated evolution between scent pads and scent patches. Given that scent patches are widespread across Lepidoptera, while scent pads are unique to Eumaeini, we believe that the evolution of patches precedes the evolution of pads in this group. Remarkably, we found (see figure below) that the rate of gain of scent pads was higher once scent patches had evolved, and we argue in the discussion that they may be ontogenetically related due to their close proximity in the wings. In other Lepidoptera, diversification of these alar androconia is related to gains and losses of new domains in critical developmental genes [6].

Thus, we believe that scent patches facilitate the evolution of scent pads, not due to a lower functional redundancy, but rather through potentially shared developmental pathways. Nevertheless, we believe that there could still be functional redundancy between these organs, since we observed that scent patches are

lost with a higher likelihood once scent pads have evolved. We have now added a sentence to this paragraph to make this point clearer (L510-512). The figure corresponds to our Fig. S18, but we have added red arrows and comments in this response to reviewers in order to highlight features of our reply.

1.9. Line 561: reversals of what?

We referred to reversals to former states as originally formulated by Louis Dollo (Gould, 1970) (hence “Dollo’s law”). We have now added to the text “reversals to former states” to enhance clarity (L519).

1.10. Line 665-667: Do you mean that the number and complexity of male secondary sexual traits “are correlated with” the intensity of sexual selection?

Thank you for bringing this to our attention. Especially in the case of secondary sexual characteristics, the differences in their numbers have been shown to be associated with the intensity of sexual selection (Jones and Ratterman, 2009; Wagner et al. 2012, Kimball et al. 2012, Simmons et al. 2017). We highlight that male secondary sexual traits might be under strong sexual and natural selection. However, the evolution of multiple and complex secondary traits involved in persuading females to mate is driven primarily by sexual selection, as noted by Darwin. In scenarios where natural selection outweighs sexual selection, due to predation risk, signal transmission and/or resource availability, this might constrain the evolution of supernumerary secondary sexual traits. We have moved this paragraph to the supplemental discussion in the Supplemental Material because of length restrictions, while we have expanded the discussion to include additional surrogates of the intensity of sexual selection, such as pheromone diversity (L570-5780).

1.11. Line 716: For Research, I suggest that the sequence data should be deposited to the NCBI Genbank so that they will have better chances to be used in the future.

We have now prepared the submission of the molecular data to NCBI and phylogenies, sequences, and phenotypic data will be available in the Dryad repository, which will be released upon acceptance (L666-L668).

Referee 2

This is an exciting manuscript reporting a higher level phylogenetic hypothesis for a species rich group of butterflies, and an analysis of how the evolution of male secondary sexual characters may have affected their diversification. I find no fault in the analyses or the interpretation of the results. A very nice study!

We thank the reviewer for these generous comments.

Referee 3

This paper resolved to explore the evolution and maintenance of male secondary sexual traits across the Lepidopteran tribe, Eumaeini. They used an AHE approach for a subset of 78 species (including 4 outgroups) to generate a phylogenomic data based on ~160k bp per species (378 loci) to generate a backbone as a constraint for full ML analyses that included additional data for 113 species based on ~11k

bp per species (13 loci). Although Eumaeini is monophyletic, the authors point out several groups are non-monophyletic, but the major clades corresponded to main biogeographic events. In addition, interestingly, the evolution of several secondary male traits are co-dependent with strong asymmetries in gains and losses.

The scope of the paper is impressive, with substantial taxa and loci representation across ~90% of the genera of the Eumaeini. Analyses selected by the authors for reconstruction of the phylogeny and phenotypic evolution seem appropriate. The phylogeny presented by the authors is also relatively well resolved with good support. Overall, I think the manuscript is well-written and the narrative is clear. However, I do have several specific comments that I have detailed below:

Specific comments

3.1. L78-80: *How are these results consistent with the trade-off between natural and sexual selection? I suggest removing this sentence because I do not see the connection in this paragraph. I.e. where are the ‘natural selection’ traits that are trading off with the sexual traits? Instead I think the authors can just proceed with “Our results illustrate...”.*

We thank the reviewer for raising this because we think it is critical to our overall argument about trade-offs. We do not differentiate between “‘natural selection’ traits that are trading off with the sexual traits”, rather we highlight that male secondary sexual traits are subject to natural and sexual selection and not only to sexual selection. We refer to the conflict between natural and sexual selection (in the Darwinian sense), in which sexual selection favors traits often exclusively expressed in one sex (typically males) that can increase mating success when there is competition, while natural selection favors economically efficient traits that can enhance viability and reproduction by improving foraging ability, predator evasion, disease resistance, etc. To clarify this, we have added a new paragraph to the Introduction (L107-L123) (see also our response to the next question). We also emphasize in the abstract and throughout the text that maintaining multiple male secondary sexual organs is likely to be costly due to the metabolic cost of producing these traits as well as the greater conspicuousness that these traits may provide to predators. If these traits are costly, they should be subject to negative natural selection, and we would expect to find negative correlations between them at a macroevolutionary level.

The large number of species in the Eumaeini has enabled us to make robust comparisons across the phylogeny, and we have indeed recovered a negative correlation between these traits. If sexual selection were the only force driving the evolution and maintenance of secondary sexual characters, we might expect to observe an accumulation of sexual characters, but instead we observe that while the common ancestor of the Eumaeini was likely to have possessed at least three androconial organs, most extant species harbor only one of these organs (Figure 2). We therefore infer from these results that maintaining these secondary traits is costly. Our findings highlight the evolutionary dynamics of these male secondary sexual organs and pave the way for future research to measure the costs of producing these traits and the strength of natural selection.

3.2. L125: *I think the introduction is a little sparse to be honest. I think authors can expand a bit more on the trade-offs component since this plays a big role in their overall narrative.*

Thank you for this suggestion. We agree and have rearranged and expanded the introduction to include a paragraph (see below) devoted to some of the assumptions implicit in our arguments about tradeoffs as described above (L107-L123).

“Multiple sexual characteristics are often costly to maintain, not only because they require the allocation of more metabolic resources, but also because they can lead to greater conspicuousness to predators [18–20]. Among a suite of males with multiple traits, selection will favor those males whose traits confer the greatest net fitness benefit because they are, for example, relatively less costly, more detectable, or more informative [21,22]. If each trait represents a major investment that is traded off against other life-history investments, it is likely that two such costly investments can induce an allocation conflict strong enough to drive a negative phenotypic correlation between the two traits [23,24]. At the microevolutionary level, there is widespread evidence of negative correlations between male traits; nonetheless, far less is known about tradeoffs between male traits at macroevolutionary timescales [24]. Given the numerous male secondary organs exhibited by Eumaeini, this tribe is an ideal group to assess this hypothesis of tradeoffs in the evolution of different male traits.”

3.3. L144-145: *I think this section could be more informative. How were the samples collected? What is the provenance and condition of samples, i.e. all samples from museum collections, in ethanol or snap frozen? All adult tissue or larvae from cultures? It will be useful to include a supplementary file detailing specifics of each sample used in the genomic study.*

We thank the reviewer for pointing this out. To provide the complete information of the voucher samples, we have added a new table including specimen data to the dryad repository and the supplementary material (Table S3), which also includes information about preservation and sampled tissue. The specimens sequenced for this study represent a combination of dry museum samples and ethanol-preserved tissues of specimens collected specifically for this study. Samples of adult butterflies were netted in the field and either papered in glassine envelopes with a silica gel desiccant or dismembered in the field, with wings stored in a glassine envelope and the body immersed in a tube of pure ethanol. In addition, we have now added a “Taxon sampling and DNA extraction” section to the Supplementary Methods, including more information about the sampling and DNA sequencing strategy.

3.4. L155: *Were whole samples (including wings) used for the extraction or just specific body parts?*

We extracted DNA from a small piece of abdominal, thoracic tissue, and/or legs removed from museum or field-collected specimens. We now provide these details in the Supplementary Methods section and in a column in the new Table S3 (see Supplementary Material).

3.5. L156: *gbiosciences.com should be replaced with the manufacturer name and country.*

Done.

3.6. L157: *What was the sequencing platform used?*

We used Illumina Hi-Seq for the 450- and 13-loci kits; and ABI Big Dye Terminator v3.1 chemistry (Applied Biosystems, Carlsbad, CA, USA) for the 5 markers. We have now added this information in the Supplementary Methods section.

3.7. L198 and 203: *Is the same dataset being referred to here?*

We thank the reviewer for pointing this out. This was a mistake since we were actually referring to the dataset with 14 loci and 191 species. It has been corrected (L198).

3.8. L233: *"We gathered additional data on morphological traits from recently published studies". Purely male sexual traits? How about other traits including overall body size or any female traits?*

We gathered additional data only for male sexual traits. Unfortunately, for Eumaeini there are no known female morphological traits that would be analogous to the male secondary sexual organs. More generally, female morphology has been poorly studied across this tribe, with the exception of the Calycopidina subtribe (Duarte and Robbins, 2010), as collections have focused on the conspicuous and colorful males. The additional data that we gathered was for males of species that were not represented in the museum collections from where we retrieved data; we now explain this in a higher detail (L237-L239). Most of the information that we obtained from the literature came from matrices of morphological characters used for cladistic analyses, where data on other traits such as body size is not available.

3.9. L245: *Missing (VHwA)*

We thank the reviewer for spotting this mistake. We have now included VHwA in the text (L247).

3.10. L246: *From each of the 819 species?*

We examined 793 species out of the 818 species included (one species was synonymized so now we have 818 instead of 819). For the remaining 25 species that we were not able to examine, we gathered data from the published literature. We have now changed this accordingly (L237-L239).

3.11. L238-245: *The section on androconial organs seem rather abrupt since nothing was mentioned in the introduction. I think the inclusion of these traits need to be justified and explained in the introduction.*

We have added a sentence in the introduction (L103-L105) to clarify that Eumaeini can possess androconial organs in addition to the three that we describe in the introduction. We illustrate their overall prevalence in Fig. 2B-C, but we have not analyzed their individual patterns of gains and/or losses, because homology cannot be assumed and they are each relatively rare, which is evident when mapped onto the phylogeny (Fig. 2C).

3.12. L299: *I think the authors need to at least include a line or two regarding the comparison between the constraint (i.e with the phylogenomic backbone) versus non-constraint approach since they mention it in the methods (L199). I did not find it referenced in the supplementary document either.*

We thank the reviewer for pointing this out. We have now included in the Supplementary Material the results of our approximately unbiased test (AU) between the best constrained and unconstrained trees, confirming that the constrained search was sensible.

3.13. L288-292: *Not convinced of why authors use Pearson momentum correlation test. Why exclude phylogenetic relatedness?*

We had phylogenetic information for 187 species (comprising nearly all the genera in the Eumaeini), allowing us to make a phylogenetic correction when analyzing this smaller dataset. When analyzing the larger morphological dataset (comprising 818 species), we did not have phylogenetic information that would enable us to correct for shared evolutionary history within genera. The most appropriate methods to analyze correlations between discrete traits rely on completely resolved phylogenies, so we therefore opted to use Pearson's correlation coefficient as a descriptive statistic showing the strength of the correlation. Nonetheless, we also explored the following alternative approach. We assumed monophyly across all genera and added species to their respective genera as unresolved polytomies. We then used this tree to calculate Pearson correlation coefficients while accounting for phylogenetic signal, using the function "corphylo" from the "ape" package (a comprehensive literature review suggested that "corphylo" is the only available method that allows unresolved polytomies). We recovered significant negative correlations for both the small and large dataset, which was consistent with what we found using only the Pearson momentum correlation.

Nevertheless, applying "corphylo" to our dataset is problematic. Firstly, it is intended for use with continuous traits because it assumes that trait evolution is given by an Ornstein-Uhlenbeck process. Secondly, while it can handle polytomies, the authors recommend it should only be used to analyze species with well-resolved phylogenetic data. In any case, because of length restrictions, we have decided to move all Pearson momentum correlation tests to the supplementary material, and we have now included a brief discussion.

3.14. L363: *Caption missing (VHwA)*

Done (L358).

3.15. L499-502: *the authors mentioned that introgression could also be a reason for the lack of resolution in the higher level phylogeny in L 317, might be good to discuss it here as well.*

We thank the reviewer for this suggestion. We have previously included a graph depicting the extent of phylogenetic discordance (see below), and we now point to this graph in the text. However, given that we lack population level data, and we cannot properly disentangle incomplete lineage sorting from introgression, we have just added a sentence acknowledging the potential role of introgression in explaining

phylogenetic incongruence between gene trees without elaborating further (L475, L479); and have included a short discussion in the Supplementary Material.

3.16. L541: "...negative tradeoff between these traits is likely to be mediated by functional redundancy..." This is a rather broad generalization. At the molecular level, the functions of these organs are more likely to be species specific, aiding in species recognition. The authors touch upon this later (L 678-680: "...it seems plausible that an interaction between sexual selection and selection for species recognition may have contributed to the establishment of early reproductive barriers..."). Selection acting on peptides and

proteins that these organs secrete likely play an important role in the usefulness of these organs and could thereby ultimately decide their gain/loss/retention in a species. These need to be considered in the discussion.

We thank the reviewer for this suggestion, and note that this was also a discussion point brought up by Reviewer 1. We have now added a paragraph discussing the role of divergent chemical evolution affecting the rate of loss of androconial organs (L570-L580):

“Additionally, multiple male traits as a mechanism aiding reproductive isolation might become less important if closely related species discriminate between the chemical signals produced by these androconia. For instance, *Heliconius* species do not vary significantly in their number of androconial organs; however, there is striking between-species variation in chemical bouquets [57,58]. These chemical blends drive female preference for conspecific males in sympatric, co-mimetic species, and seem to evolve rapidly, with pheromone gains and losses occurring frequently across the *Heliconius* phylogeny [58–60]. Hence, rapid divergent chemical evolution might render the loss of multiple secondary sexual characters more likely. Currently, the chemical ecology of individual species of Eumaeini remains largely unknown, but the macroevolutionary patterns shown here suggest that this would be a productive avenue for future research.”

We highlight the role of functional redundancy because implicit in the idea of tradeoffs between different androconial organs is the idea that they must involve a cost to the male; otherwise, we would not expect to see a macroevolutionary pattern in which the presence of one organ increases the likelihood of loss of the second organ. We believe it is significant that we see a trade-off rather than an unbounded accumulation of secondary sexual traits over time. However, it is true that selection could act to modify the function of these traits in a way that might reduce any redundancies, so it would be especially interesting to study those taxa with an unusually high number of androconial organs to see whether these glands have a more specialized function and/or the species that have them also share specific life history traits. We have also added a paragraph about this in the Introduction (L107-L123).

Language comments

3.18. *Overall the language structure fluctuates between passive and active voice. I urge the authors to stick to one.*

We thank the reviewer for pointing this out. Different authors have different preferences, making it difficult to maintain a single voice, but we have gone over the final draft carefully with this in mind to try to make it as consistent as possible.

3.17. *L69: “We examined specimens from 819 of the 1,096 described species (75%) and documented that male secondary sexual traits are present in 91% of the species in our dataset.” Not accurate to say 91% of known species if only examined a subset.*

We thank the reviewer for pointing out this mistake; we have now changed it to be: “males of 91% of surveyed species” (L59).

3.18. L71: *Quotations not necessary for sexual traits.*

We have now removed the quotation marks.

3.19. L72: “... *were likely present in...*”

We have now changed it to be “were probably present in” (L60). (We did not use the word “likely” simply to avoid word repetition).

References

- dos Reis M, Yang Z. 2019 Bayesian Molecular Clock Dating Using Genome-Scale Datasets. In *Evolutionary Genomics: Statistical and Computational Methods* (ed M Anisimova), pp. 309–330. New York, NY: Springer. (doi:10.1007/978-1-4939-9074-0_10)
- Duarte M, Robbins RK. 2010 Description and phylogenetic analysis of the Calycopidina (Lepidoptera, Lycaenidae, Theclinae, Eumaeini): a subtribe of detritivores. *Rev. Bras. entomol.* 54, 45–65. (doi:10.1590/S0085-56262010000100006)
- Espeland M et al. 2018 A comprehensive and dated phylogenomic analysis of butterflies. *Current Biology* 28, 770–778.e5. (doi:10.1016/j.cub.2018.01.061)
- Gould SJ. 1970 Dollo on Dollo’s law: irreversibility and the status of evolutionary laws. *J Hist Biol* 3, 189–212. (doi:10.1007/BF00137351)
- Jones AG, Ratterman NL. 2009 Mate choice and sexual selection: What have we learned since Darwin? *PNAS* 106, 10001–10008. (doi:10.1073/pnas.0901129106)
- Kimball RT, Mary CMS, Braun EL. 2011 A macroevolutionary perspective on multiple sexual traits in the Phasianidae (Galliformes). *Int. J. Evol. Biol.* 2011, 423938. (doi:10.4061/2011/423938)
- Martins ARP, Duarte M, Robbins RK. 2019 Phylogenetic classification of the *Atlides* section of the Eumaeini (Lepidoptera, Lycaenidae). *Zootaxa* 4563, 119. (doi:10.11646/zootaxa.4563.1.6)
- McGuire G, Denham MC, Balding DJ. 2001 Models of Sequence Evolution for DNA Sequences Containing Gaps. *Molecular Biology and Evolution* 18, 481–490. (doi:10.1093/oxfordjournals.molbev.a003827)
- Pagel M. 1994 Detecting correlated evolution on phylogenies: a general method for the comparative analysis of discrete characters. *Proceedings of the Royal Society of London. Series B: Biological Sciences* 255, 37–45. (doi:10.1098/rspb.1994.0006)
- Plummer M, Best N, Cowles K, Vines K. 2006 CODA: Convergence Diagnosis and Output Analysis for MCMC. *R News* 6, 7–11.
- Prakash A, Monteiro A. 2020 Doublesex mediates the development of sex-specific pheromone organs in *Bicyclus* butterflies via multiple mechanisms. *Mol Biol Evol* 37, 1694–1707. (doi:10.1093/molbev/msaa039)
- Simmons LW, Lüpold S, Fitzpatrick JL. 2017 Evolutionary Trade-Off between Secondary Sexual Traits and Ejaculates. *Trends Ecol. Evol.* 32, 964–976. (doi:10.1016/j.tree.2017.09.011)
- Wagner CE, Harmon LJ, Seehausen O. 2012 Ecological opportunity and sexual selection together predict adaptive radiation. *Nature* 487, 366–369. (doi:10.1038/nature11144)